

# Using *in-situ* environmental DNA sampling to detect the invasive New Zealand Mud Snail (*Potamopyrgus antipodarum*) in freshwaters

Jake J. Ponce[1], Ivan Arismendi[1] and Austen Thomas[2]

[1] Department of Fisheries, Wildlife, and Conservation Sciences, Oregon State University, Corvallis, OR, United States of America
[2] Molecular Division, Smith-Root, Inc., Vancouver, WA, United States of America

## ABSTRACT

Environmental DNA (eDNA) detection of aquatic invasive species is currently at the forefront of aquatic conservation efforts because the methodology provides a cost effective and sensitive means to detect animals at low densities. Developments in eDNA technologies have improved detection probabilities for rare, indicator, and invasive species over the past decade. However, standard lab analysis can take days or weeks before results are available and is prohibitive when rapid management decisions are required for mitigation. Here, we investigated the performance of a real-time quantitative PCR system for on-site eDNA detection of New Zealand mud snails (*Potamopyrgus antipodarum*). Six sites in western Washington, USA were sampled using the rapid eDNA technique and traditional methods, with five samples per site. On-site eDNA detection of mud snails resulted in a 10% increase in positive sites (16/30 = 53% positive) relative to visual surveys (13/30 = 43% positive). In addition, positive associations were observed between mud snail eDNA concentration (eDNA copies per reaction) and the number of mud snail individuals at each site ($R^2 = 0.78$). We show that the rapid on-site eDNA technology can be effective for detection and quantification of New Zealand mud snails in freshwaters. This on-site eDNA detection approach could possibly be used to initiate management protocols that allow for more rapid responses during the onset of biological invasions.

# INTRODUCTION

Aquatic invasive species can dramatically alter native communities by reducing biodiversity and changing ecological processes (*Alonso & Castro-Diez, 2008*). For example, an invasive mollusk the zebra mussel (*Dreissena polymorphia*), has been known to modify supporting and provisioning services in aquatic ecosystems through alteration of water quality and bioaccumulation of primary production (*Vila et al., 2010*). In addition, the red swamp crayfish (*Procambarus clarkii*) one of Europe's top 100 worst invaders has had substantial detrimental impacts on invaded freshwater ecosystems through competition with native species, predation, and alteration of ecosystem services (*Lodge et al., 2012*;

Corresponding author
Jake J. Ponce, poncejake@gmail.com

*Treguier et al., 2014*). Collectively, invasive species can have a large impact on our local and global economy with the United States alone spending roughly $131-185 billion dollars per year on invasive species issues (*Marbuah, Gren & McKie, 2014*).

Monitoring for invasive species at all levels of the invasion process (*i.e.,* introduction, establishment, and spread; *Kolar & Lodge, 2001*) can be cost prohibitive. Management teams must balance between the high costs of surveillance and the potential eradication effort that is necessary if there is a failure to detect introduced animals at low abundances (*Hayes et al., 2005*). Specifically, aquatic invertebrates are notoriously difficult to inventory using traditional sampling methods due to their small size, low population densities, patchy distributions, and the complexity in the use of habitat during different life stages (*Barbour et al., 1999*). As a result, many bioassessment methods involving identification of taxa only to the genus or family level may be inferior to information collected at the species level when evaluating aquatic management strategies for endangered or non-native species (*Mächler, Deiner & Altermatt, 2014*). Moreover, traditional sampling methods (*i.e.,* Hess sampler or D-frame kick nets) requires long processing time, can cause injury to target and non-target species, and prey species are readily exposed to predators after collection (*Snyder, 2004*; *Ghani et al., 2016*). Conducting these traditional sampling methods can increase the risk of spreading invasions through the transport of sampling gear that have not been properly decontaminated (*Bersine et al., 2008*; *Veldhoen et al., 2016*).

Rapid technological advancements including environmental DNA (eDNA) detection have led to increases in sampling sensitivity at a lower cost than traditional sampling (*Evans et al., 2017*). Assessment of biodiversity using eDNA relies on a molecular workflow comprised of several steps including the capture, extraction, and identification of an organism's DNA from environmental samples such as soil or water (*Huerlimann et al., 2020*). eDNA methods were first applied in sediments, revealing DNA from extinct and extant animals and plants, and since then, they have been used to detect taxa from terrestrial and aquatic environments (*Thomsen & Willerslev, 2015*; *Coble et al., 2019*). eDNA provides an option for detection of species without physical capture or visual confirmation (*Davy, Kidd & Wilson, 2015*). Due to rapid emergence of eDNA for detection of flora and fauna, it has generated interest among fisheries and other natural resource managers seeking cost-effective tools for aquatic species inventory and monitoring (*Pilliod et al., 2013*; *Coble et al., 2019*; *Penaluna et al., 2021*).

Currently, there is not a standardized eDNA sampling protocol for freshwater ecosystems, but the overall sequence of steps includes water collection, preservation, and laboratory analysis (*Ruppert, Kline & Rahman, 2019*). Further, research suggests that sampling decisions made during water collection (*i.e.,* volume of water) and handling steps (*i.e.,* on-site *vs.* lab filtration) may influence detection (*Sepulveda et al., 2019*; *Curtis, Larson & David, 2021*). *Sales et al. (2019)* found that samples collected in containers, stored on ice, and then transported back to a lab for filtration yielded higher target species copy numbers than the use of a chemical preservation method. However, a better approach to maximize DNA concentration is to filter water samples on-site. Indeed, *Yamanaka et al. (2016)* show that filtering on-site compared to transporting water back to the lab on ice for filtration, yielded 2.21 times more DNA suggesting that immediate filtration could be

especially relevant for species that are rare and in low abundances. While eDNA has several advantages compared to traditional sampling methods, eDNA sampling still requires days, weeks, or even months to receive results depending on the quantity of samples, shipping logistics, and laboratory lead times. Time-sensitive studies or management requirements may have the need for rapid on-site sampling detection processes which could alleviate these complications.

Here, we investigate the eDNA field-based platform demonstrated by *Thomas et al. (2019)* with the Smith-Root eDNA-Sampler and the Biomeme field-portable qPCR thermocycler (hereafter "eDNA field platform") technologies to determine if this method is suitable to provide rapid detection of the non-native New Zealand mud snail (*Potamopyrgus antipodarum*) in freshwaters. This species is native to New Zealand's freshwater lakes and streams and is currently an invasive species to North America (*Hall, Tank & Dybdahl, 2003*). New Zealand mud snails (hereafter "mud snails") have been documented to exceed densities greater than 400,000 snails per square meter around Yellowstone National Park (*Hall, Tank & Dybdahl, 2003*) and were found a decade later in the estuary of the Columbia River (*Zaranko, Farara & Thompson, 1997*). Mud snails have continued to spread from the West Coast (*Benson et al., 2019*) and throughout the world (*Schreiber et al., 1998*). In this study, we validate the eDNA field platform with traditional field sampling methods in search of mud snails in western Washington, USA. In addition, we explore biologically relevant covariates including snail densities, ambient water temperature, and ambient water conductivity and their association with the relative abundance of mud snail eDNA from water samples. Our study provides insights into the use of real-time sampling protocols to quickly inform natural resource managers about early detection of biological invasions.

## MATERIALS & METHODS

### The New Zealand mud snail

Mud snails are herbivore/detritivores and were first found in North America in the Middle Snake River in Idaho in 1987 and were thought to have escaped from a fish farm (*Bowler, 1991*; *Hall, Tank & Dybdahl, 2003*). Mud snails occupy a wide range of habitats, are believed to be a successful invader due to their unique biological features that allow for easy transport to new water bodies and have adaptable reproductive capabilities that have been documented throughout the world (*Schreiber et al., 1998*). A single female mud snail can produce up to 120 embryos and bear approximately 70 live individuals every three months (*Cheng & LeClair, 2011*). In their native range, these small gastropods (adults <five mm in length) may reproduce sexually or by parthenogenesis, whereas the non-native populations in North America are known to be all-female clones (*Hall, Dybdahl & VanderLoop, 2006*). In addition, a shift in nutrient fluxes such as carbon or phosphorus caused by high density populations of mud snails could have a large influence on primary production rates that can affect native grazers and ecosystem functions (*Tibbets et al., 2010*). Grazing trials conducted with mud snails and native grazers concluded that mud snails were more efficient grazers (*Larson & Black, 2016*).

With the ability to generate large populations, mud snails can invade ecosystems through varying dispersal methods and have been known to extend their range by over

640 km along the Snake River, Idaho in as little as ten years (*Zaranko, Farara & Thompson, 1997*). Currently, there is no available method for *in situ* eradication of mud snails once they invade without harming the entire ecosystem (*Schisler, Vieira & Walker, 2008*). Minimizing the spread of mud snails include a multitude of efforts and management plans such as educational outreach efforts, chemical application, and mechanical application such as freezing and drying of all encountered field equipment (*Aquatic Nuisance Species Task Force, 2006*).

## Study sites

We targeted mud snails in western Washington that encompassed a wide range of watersheds and environmental factors to gather a greater representation and variability when comparing the eDNA platform to traditional sampling methods. Our study area consisted of six freshwater streams including Burnt Bridge Creek, Columbia River (Kalama River, WA confluence), Capitol Lake (Deschutes River, WA confluence), May Creek, McAleer Creek, and Thornton Creek (Fig. 1). All streams were traditionally surveyed in the past from either Washington Department of Fish and Wildlife or the United States Geological Survey and were confirmed to be positive sites of mud snails with varying relative abundances. Streams were selected based on positive identifications of mud snails, ease of access, and quantity levels of mud snails to compare sites from trace amounts to well established sites.

Burnt Bridge Creek, Capitol Lake, and Thornton Creek had a vast amount of aquatic vegetation and soft soil substrates, whereas the Columbia River, May Creek, and McAleer Creek encompassed sand or small cobble sediments. Streams were sampled in May of 2019 to increase probability of detection as seasonal activities and warming temperature conditions can create a reproduction event for mud snails (*Schreiber et al. , 1998*; *Ardura et al., 2015*). At each stream, sampling occurred along a total length of 40-meters with one eDNA and traditional sample taken every 10-meters. Water samples were obtained first to minimize any contamination from traditional equipment and all samples started from downstream to upstream direction. Additionally, each sample site was geo-referenced using the internal GPS in the eDNA sampler. Water conductivity and water temperature was also measured *in situ* using a Hannah Primo 5 conductivity meter and HoneForest TDS/Temp meter, respectively.

## Water sampling

The eDNA field platform incorporates a backpack water pump system and a portable handheld qPCR device that allows for field sampling and processing within one hour. The eDNA water sampler technology monitors flow rate, pressure, and volume filtered that can help the user adapt to the environmental conditions to receive optimal filtration results. A full detail of the eDNA sampler setup and workflow is demonstrated by *Thomas et al. (2018)*. The Biomeme system incorporates a qPCR handheld device that has field ready kits for water samples that come with all the necessary chemistries to extract and run samples for species-specific detections. Further details of the protocols used for both systems are demonstrated by *Thomas et al. (2019)*.

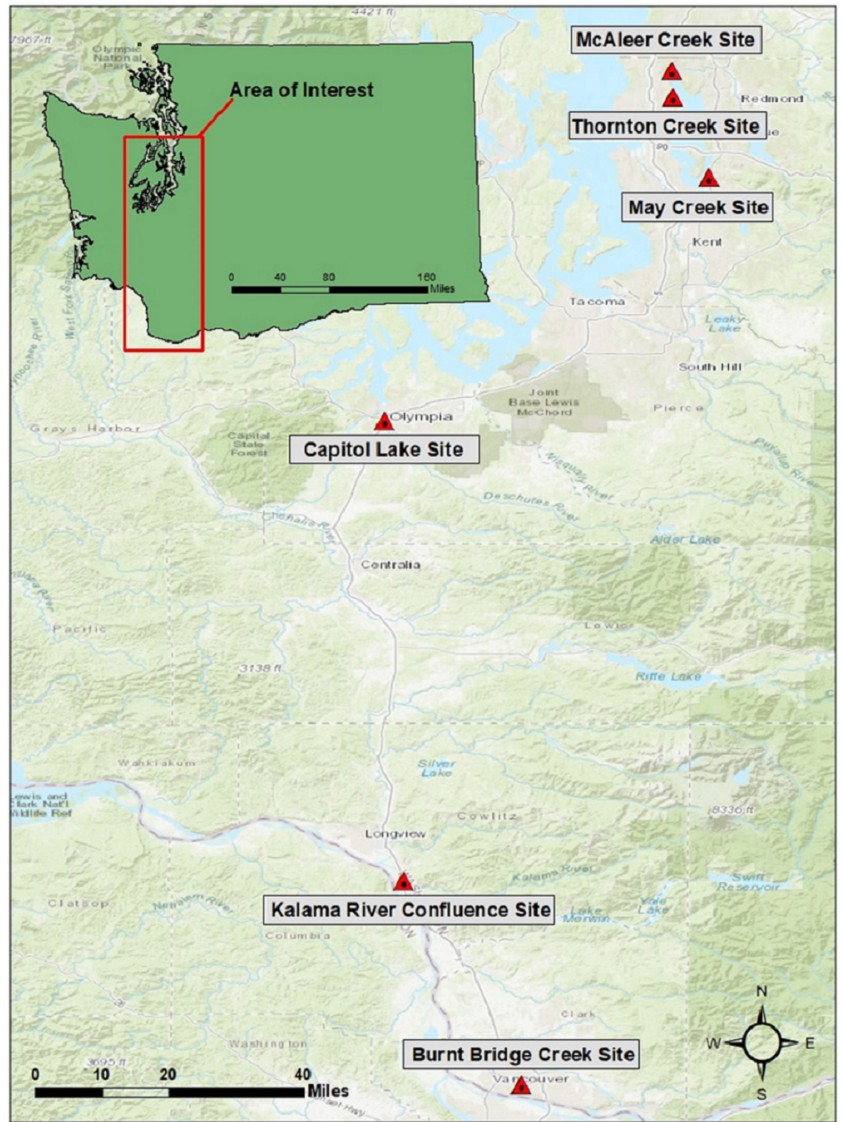

**Figure 1** **Sampling area.** Map including the six sampling sites in Washington State, United States.

We conducted water surface point sampling using the telescoping pole provided to decrease potential contamination from equipment by extending the filter away from the shore. Preliminary sampling at two sites concluded that filters were clogging between 0.75 L and 2 L, thus samples were standardized to 1L per sample with 1 L/min flow rate and 12 PSI for pressure. We used single-use filters designed for eDNA water sampling, which are 47 mm in diameter and 5.0 μm in pore size made of polyethersulfone filter paper (Sterlitech Corporation). This pump involves a negative pressure system that allows large particulates to descend through the six-inch snorkel tube that is within the water column. After filtration, the filter was kept in the filter housing and was placed back in the original packaging until all samples were completed at each site (~20 min). We changed nitrile

gloves between each eDNA sample filtration to prevent potential cross-contamination. Once all five water samples were obtained, a negative control was filtered using distilled water. These efforts resulted in six samples per stream and a total of 36 samples including field blanks. Upon completion of all six field filtrations per stream, samples were taken back to the vehicle where immediate DNA extraction began using the Biomeme extraction kit.

## Mobile qPCR

The portable real-time qPCR thermocycler utilizes shelf-stable ready to go species-specific assay strips and a 5-min DNA extraction kit. The thermocycler has the capability to analyze three DNA samples at a time and in this study, we had one qPCR replicate per water sample. Due to the limited number of wells within this machine we did not include a no-template control, but we did include a field blank at each stream to indicate any cross-contamination issues. The machine has two light channels (FAM and Cy5) and incorporates an internal positive control (IPC) that helps to determine potential inhibition. A standard curve was created by diluting a gBlock (IDT) for mud snails by running four dilutions in the wells comprising of 10, 100, 1,000, and 10,000 copies (*Thomas et al., 2018*). The limit of detection (LOD) was 10 copies per reaction and the limit of quantification (LOQ) was 50 copies per reaction.

Understanding known concentrations can help estimate target DNA concentration values when running unknown samples compared with quantification cycles. We used the mud snail assay, including primer and probe, from *Goldberg et al. (2013)* with a compatible mastermix that was then ported into the custom pre-loaded freeze-dried pellets that are shelf stable, individually sealed, and ready to be saturated. The M1-field test kit comes with multiple buffers and chemical solutions where the DNA is bound to a column syringe and is then purified with the various reagents resulting in DNA that is ready to be added to the freeze-dried pellets. We followed the manufacturers recommendations for this process, except we added a settling step after the initial lysis buffer stage to help with any inhibition issues and clogging within the syringe. After rinsing the filter with the lysis buffer, the material was allowed to settle for 15 min and then the supernatant liquid was transferred to a new 5 ml vial for the remaining extraction steps. The final step was to use the 20 μL micropipette that was included in the kit to then take the purified DNA and transfer to the lyophilized wells to re-hydrate before placing into the machine. We changed nitrile gloves between each DNA extraction to decrease the potential for cross-contamination. Additional details of this process can be found in *Thomas et al. (2019)*.

## Traditional field sampling

Each sample site was evaluated by traditional sampling with the same sampling regime as eDNA sampling that incorporated five total samples per location at 10-m increments (for reference see Fig. 2). Traditional sampling of mud snails was conducted after water samples were completed to compare detected and not detected eDNA data to physically identified mud snails to gather an estimated site abundance. At each site, we collected substrate or vegetation depending on the habitat of approximately one square meter by

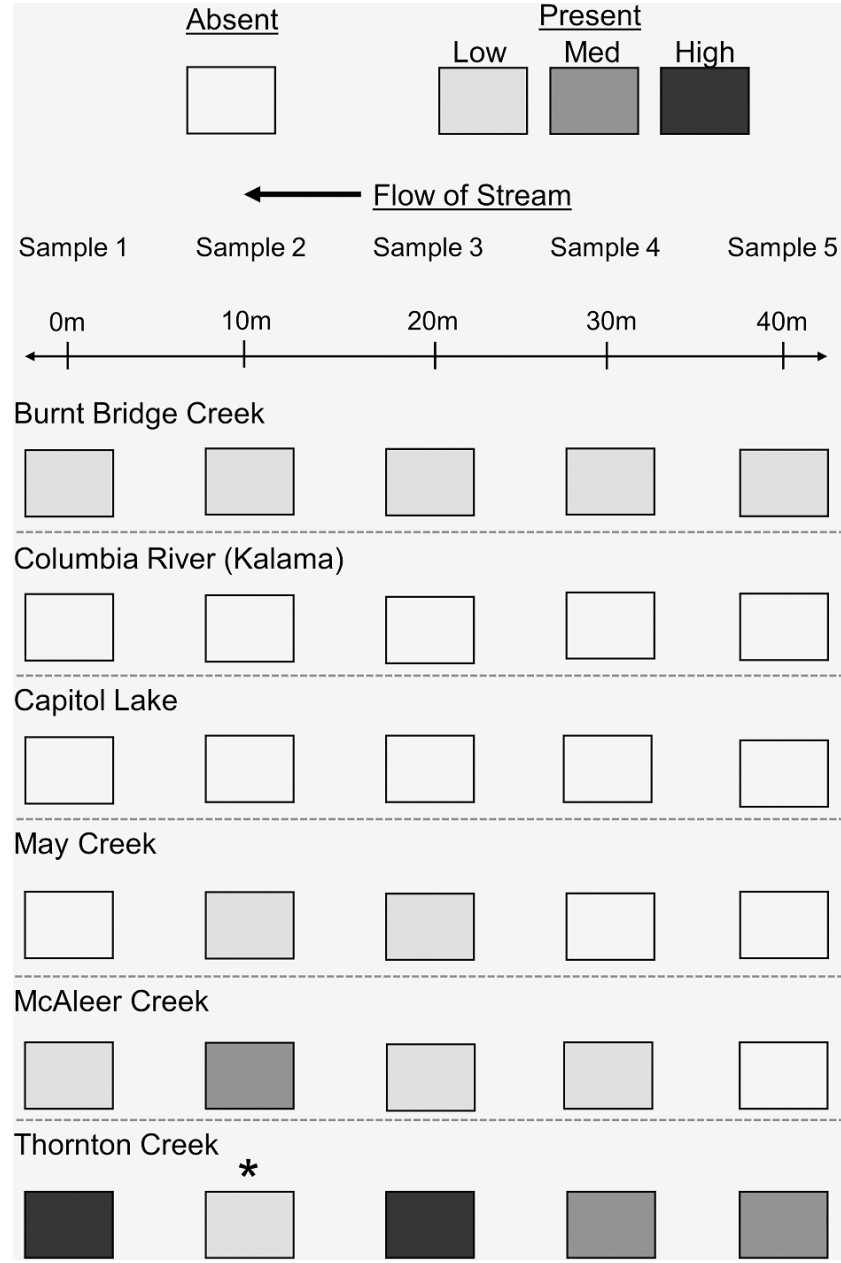

**Figure 2 Summary of eDNA sampling array per site.** Target eDNA copy numbers per reaction is represented by Low = 0–133, Med = 134–266, and High > 266. The asterisk denotes PCR inhibition.

either a D-frame kick net or shovel from the same location where the eDNA sample was taken and placed the material into a 20 L bucket. To standardize the effort, we examined the material for 10 min. at each sample site by sifting the material over a number 18 sieve (1.0 mm diameter mesh). Any snails that could not be immediately identified were further examined after 10 min. of sampling effort. We removed captured mud snails from the rivers and took them to the landfill.
**Table 1** Summary of eDNA sample detections and traditional sampling detections ($n = 30$) at multiple sites in Washington.

| Location | Total # of snails | Snail capture sites (%) | eDNA detected sites (%) | eDNA detected sites where snails were captured (%) |
|---|---|---|---|---|
| Burnt Bridge Creek | 29 | 5/5 (100) | 5/5 (100) | 5/5 (100) |
| Columbia River (Kalama) | 0 | 0/5 (0) | 0/5 (0) | N/A |
| Capitol Lake | 0 | 0/5 (0) | 0/5 (0) | N/A |
| May Creek | 0 | 0/5 (0) | 2/5 (40) | 0/2 (0) |
| McAleer Creek | 3 | 3/5 (60) | 4/5 (80) | 2/4 (50) |
| Thornton Creek | 165 | 5/5 (100) | 5/5 (100) | 5/5 (100) |
| Total | 197 | 13/30 (43) | 16/30 (53) | 12/16 (75) |

## Statistical analysis

We evaluated the associations between eDNA starting quantity (SQ) or target DNA concentration and environmental covariates including the visual number of mud snails captured, ambient water conductivity, and ambient water temperature using a simple linear regression. Due to limited sample sizes, we combined and averaged values from each stream. Data points were identified for comparisons if the SQ value was greater than the LOQ and positive detections were counted if the SQ was greater than the LOD. To facilitate the visualization of our findings (see raw data as supplement), mud snail DNA copy numbers were grouped and summarized as low (<133 copies), medium (134–266 copies), and high (>266 copies). Streams resulting in non-detections for eDNA samples were removed from the analysis. All statistical analyses were conducted using RStudio (2019).

## RESULTS

The eDNA field platform successfully detected and quantified DNA concentrations of mud snails from water samples. We were unable to detect mud snails from either sampling method for all six sampling sites in western Washington that were previously surveyed from other agencies who confirmed to have at least one positive snail detection from traditional sampling. At the most upstream sample site in McAleer Creek, we were able to visually identify one mud snail, but did not detect it with the eDNA field platform. However, we detected eDNA at the four samples downstream indicating the importance of both DNA transport and patchy distribution of snails across sample sites. While we only had one field replicate per sample, we included multiple samples per site to increase the likelihood of a positive detection. Inhibition was evaluated by the lack of amplification with the IPC and we observed one sample exhibiting inhibition equaling a 3% ($n = 30$) overall field sample inhibition rate (Fig. 2). All control samples resulted in no amplification indicating our contamination sampling protocols were effective in eliminating DNA from other sources. We found that using the eDNA field platform equated to a 10% increase in detection compared to traditional sampling (Table 1).

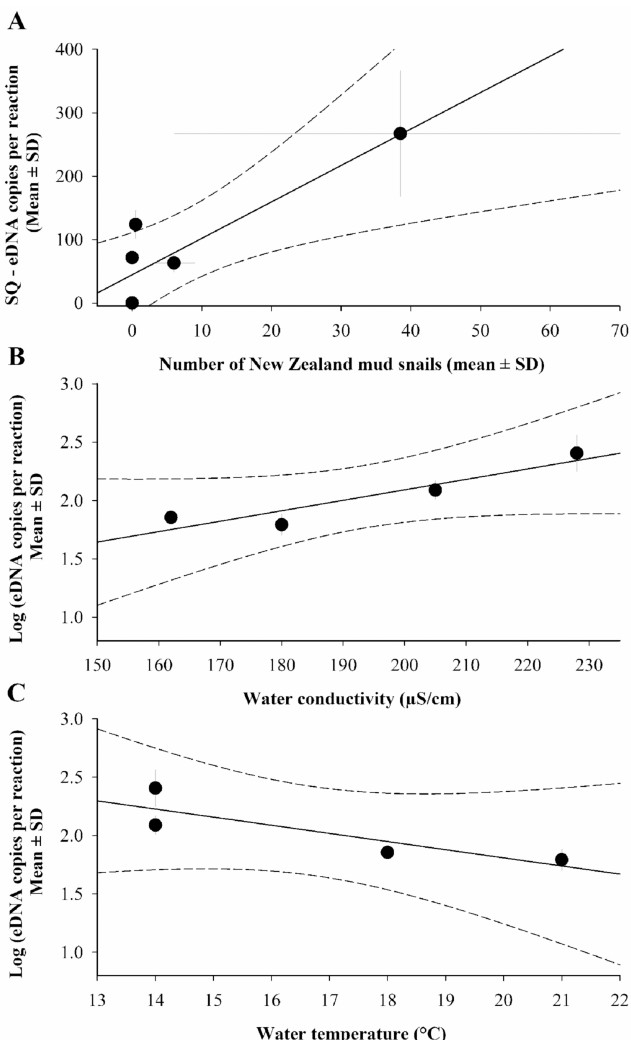

**Figure 3  Relationship between New Zealand mud snails (*Potamopyrgus antipodarum*) eDNA abundance (copies per reactions) and environmental covariates at the six streams sampled.** (A) Average number of captured mud snails ($R^2 = 0.78$); (B) ambient water conductivity ($\mu$S/cm) –2 streams had no eDNA detections in all samples (individuals per m$^2$) ($R^2 = 0.87$); (C) ambient water temperature (°C) –2 streams had no eDNA detections in all samples ($R^2 = 0.73$). Dotted lines indicate 95% confidence bands.

For sites with agreement, the results indicate a positive relationship between mud snail DNA copy number per reaction and relative number of mud snail individuals ($R^2 = 0.78$, Fig. 3A). Similarly, ambient water conductivity showed a positive relationship between the detection of mud snails and eDNA ($R^2 = 0.87$, Fig. 3B), and water temperature showed a strong association with snail DNA copy number per reaction ($R^2 = 0.73$, Fig. 3C).

Mud snail relative abundance was evaluated at sites with positive detections and ordinal data (Fig. 2). Mud snail DNA concentration per reaction was categorized with low (0–133), medium (134–266), and high (>266) values. Thornton Creek was the only site to observe all three eDNA relative abundance levels within the 40-m sampling distance along the shore

and had the highest mud snail DNA concentrations. May Creek had two positive detections within the middle of the sampling section indicating the unique patchy distribution that mud snails can exhibit in small streams.

## DISCUSSION

Here, we demonstrate that the eDNA field platform can detect mud snails in rivers of western Washington. At every site where we collected mud snails with traditional sampling, we also detected them with eDNA, except in one sample at McAleer Creek probably owing to low abundances (1.0 snail/m$^2$). New Zealand mud snails can bury themselves within sediments (*Duft et al., 2003*), making it difficult for them to shed enough DNA for detection with eDNA methods.

Our findings demonstrate the patchy distribution that mud snails can exhibit in the environment and can make surveys whether traditional or eDNA sampling challenging. Additional water volumes or the use of more replicates can potentially solve this problem to maximize DNA detection. In our case, preliminary filtering at Burnt Bridge Creek under high turbidity conditions clogged the filter around 1.0 L. However, the eDNA field platform detected 20% more positive sites at McAleer Creek even lacking the one positive detection where we encountered one mud snail. Other field studies that compare traditional sampling techniques to eDNA methods report positive detections at the same site along with visual surveying (*Dejean et al., 2012*; *Takahara, Minamoto & Doi, 2013*). In some cases, traditional fish snorkeling can be more effective than eDNA, but this could be due to insufficient water volumes sampled, low densities of the target species, mobility of fish in lotic environments, and eDNA dilution from flowing water (*Ulibarri et al., 2017*). In other cases, eDNA seems to be more effective in detecting the end of fish distribution in headwater streams than traditional electrofishing techniques (*Penaluna et al., 2021*). Similarly, our study shows that eDNA has the potential to provide more positive detections of mud snails than traditional sampling.

The positive association between mud snail DNA SQ and the relative abundance of mud snail animals from traditional sampling is promising. However, there is a lack of literature regarding mud snail abundance compared to eDNA except for *Goldberg et al. (2013)* which evaluated eDNA concentrations and the relationships between mud snail densities in an aquarium and field applications. Similar research has shown eDNA relationships are stronger overall in laboratory settings compared to field applications (*Yates, Fraser & Derry, 2019*). Though, relationships between relative abundances calculated by eDNA and traditional approaches have been documented for fish, amphibians, and freshwater mollusks (*Takahara et al., 2012*; *Thomsen et al., 2012*; *Baldigo et al., 2017*; *Iwai, Yasumiba & Takahara, 2019*; *Stoeckle et al., 2020*). For example, *Pilliod et al. (2013)* showed positive relationships between eDNA and traditional methods for both Rocky Mountain Tailed Frogs (*Ascaphus montanus*) and Idaho Giant Salamanders (*Dicamptodon aterrimus*). In another case, there was no relationship between abundances of eDNA and traditional methods for the Eastern Hellbender *Cryptobranchus alleganiensis alleganiensis* (*Spear et al., 2015*).

We found a positive relationship between water conductivity and mud snail DNA concentrations. This finding can be explained by gastropods generally requiring habitat with high calcium ions, pH, and conductivity (*Dillon, 2000*). Conversely, water temperature conditions resulted in a negative relationship. Temperature conditions from our study sites include suitable ranges for mud snails (*Dybdahl & Kane, 2005*). However, the variability of snail reproduction affected by temperature and eDNA degradation due to warmer waters at two sites may affect our findings (*Strickler, Fremier & Goldberg, 2015*); (*De Souza et al., 2016*).

The use of eDNA as well as our approach have some limitations to consider. For example, PCR inhibition occurs under high concentration of organic compounds that includes humic and fulvic acids (*Wetzel, 1992*; *Albers et al., 2013*; *McKee, Spear & Pierson, 2015*). eDNA can also be degraded by distance traveled, pH, and UV radiation inducing false negatives (*Jane et al., 2015*; *Klymus et al., 2015*; *Strickler, Fremier & Goldberg, 2015*). It is recommended that for early detection surveys, additional replicates are included when results are positive to confirm presence or when false negatives may occur due to low abundances (*Rees et al., 2014*; *Goldberg et al., 2016*; *Clusa et al., 2017*). In our study, the use of one replicate is a limitation, however, our approach incorporates additional samples within the same proximity (40 m) which increases our chances of detection given the patchy distribution of mud snails. We encourage further research to consider mud snail-specific study designs before any broader survey implementation. We did our best to minimize our study limitations by including filtration and DNA extraction *in situ*. Specifically, we decreased the prevalence of inhibitors by using a negative pressure pump system that helps to reduce large particulates on the filter membrane. The 3% inhibition rate in our study is comparable to *Nguyen et al. (2018)* and lower than others (*Sepulveda et al., 2018*; *Thomas et al., 2019*). We incorporated a settling step to decrease PCR inhibition over space, however, further refinement may be needed pending different environmental conditions and or a pilot study to confirm sampling protocols.

Early detection of species in low numbers or that are elusive such as non-native species using eDNA is critical for managers to respond with the best strategy to prevent environmental degradation (*Pluess et al., 2012*; *Anglès d'Auriac et al., 2019*; *Penaluna et al., 2021*). There are still many questions about best practices of using eDNA to detect invasive species in freshwaters. For example, eDNA production rates can vary among differing species, densities, age classes, and seasonal changes (*Maruyama et al., 2014*; *Sassoubre et al., 2016*). In streams, transport, settling, and dilution can affect detection probabilities and it is documented that eDNA detections can occur up to 12 km downstream (*Deiner & Altermatt, 2014*). Understanding the habitat and species-specific environmental traits where eDNA sampling is conducted will help natural resource managers choose specific sampling strategies to be deployed to increase success in early detection of invasive species.

## CONCLUSIONS

We show that the eDNA field platform used here can detect mud snails within a rapid response time (~1 h). This equipment is less expensive for long-term management

practices, does not require lab-based equipment or specialized personnel to manipulate samples and report results. Utilizing the eDNA field platform is a promising tool in natural resource managers tool bags for battling the forefront of aquatic invasions.

## ACKNOWLEDGEMENTS

Jesse Schultz from Washington Department of Fish and Wildlife for providing the locations of the animals and Mieke Sinnesael from Biomeme, Inc. for her input and technical help with the qPCR equipment. Drew Carey for his assistance with GIS. Two anonymous reviewers, Shaun Wilkinson, and the Associated Editor, Xavier Pochon, provided excellent suggestions to improve our final version of this manuscript.

### Funding
Funding was provided by Smith-Root. The funders had no role in study design, data collection and analysis, decision to publish, or preparation of the manuscript.

### Grant Disclosures
The following grant information was disclosed by the authors:
Smith-Root.

### Competing Interests
Austen Thomas and Jake Ponce are employed by Smith-Root, Inc. However, none of the authors will directly benefit from the publication of this article.

### Author Contributions
- Jake J. Ponce conceived and designed the experiments, performed the experiments, analyzed the data, prepared figures and/or tables, authored or reviewed drafts of the paper, and approved the final draft.
- Ivan Arismendi conceived and designed the experiments, analyzed the data, prepared figures and/or tables, authored or reviewed drafts of the paper, and approved the final draft.
- Austen Thomas conceived and designed the experiments, performed the experiments, analyzed the data, authored or reviewed drafts of the paper, and approved the final draft.

### Field Study Permissions
The following information was supplied relating to field study approvals (i.e., approving body and any reference numbers):
Sampling sites were at public locations and filtering water did not require a permit.

### Data Availability
Data are available in the Supplementary File.

## Supplemental Information

Supplemental information for this article can be found online at http://dx.doi.org/10.7717/peerj.11835#supplemental-information.

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
