# Peer review of "Using in-situ environmental DNA sampling to detect the invasive New Zealand Mud Snail (Potamopyrgus antipodarum) in freshwaters"

_PeerJ, doi:10.7717/peerj.11835_

## Round 0.1 · original submission · Major Revisions

Dear Jake and co-authors,

I have received three, rather contrasting, independent reviews of your study. While all reviewers clearly recognised the quality/importance of your work, two reviewers have raised a number of issues that will need to be addressed in your revised manuscript. In particular, reviewer#1 recommended rejection in large part due to the experimental design that they claimed not appropriate to support your conclusions. I noted, however, that the other two reviewers (both established eDNA experts) were very supportive of this study and did not find that the experimental design was fundamentally flawed. Your revision is an opportunity to clarify the issues raised by reviewer#1, both in your rebuttal and manuscript. I will send your revised manuscript back to at least two reviewers for final assessment.

Overall, the reviewers have provided you with excellent suggestions on how to improve the manuscript, and I will be looking forward to receiving your revised manuscript along with a point-by-point response to the reviewers' comments.

With warm regards,
Xavier

Reviewer 1 ·

Basic reporting

The language is mostly clear, with a few sections missing potentially helpful details.
Most importantly, I worry about the literature references: Several milestone reviews on eDNA research such as Ruppert, Kline & Rahman (2019) or Huerlimann et al. (2020) have not been cited (the reviewer has not been involved in the mentioned publications); instead, some references do not contain the information they are supposed to. For example, in ll. 92-93, Lodge et al. do not refer to any cost-associated advantages of eDNA research. In addition, the citation of Machler et al. in ll. 83-85 is very close to a copy of the respective paragraph in the paper itself.
While the manuscript adheres to the professional article structure, there are worries about the depths of its analyses, what is reflected in the figures and tables (see below). The raw data is available. The hypotheses are self-contained, but results are interpreted wrongly.

Experimental design

While the research question is clearly stated, the experimental design suffers from:
(1) low sample size; (2) lack of replication per site (which is stated by the authors themselves and could have easily been included; spatial replication along a transect is not replication per site); (3) no impact of water volume on detectability is assessed (while many other eDNA studies use 1L, this manuscript focuses on the detection rate of one specific elusive species, so the volume could have been optimised); (4) while the design involves a river gradient, this spatial dependency is not being taken into account later on.
The methods are described in enough detail, whereas the co-author's previous papers were necessary to understand the entire context of the work.

Validity of the findings

The findings of the manuscript do not agree with the data.
The Fisher's exact test that is applied to compare traditional and eDNA surveying shows that there is NO significant difference between both methods based on the respective detection rates. The alternative hypothesis of a Fisher's exact test is that there is some sort of association between two different variables, which is the case in this study since for most of the sites eDNA and traditional surveys agree on the presence/absence of the species. In general, the authors describe that traditional methods detected 13 sites to be positive and eDNA methods detected 18 sites to be positive, so only 5 sites were only detected as positive by eDNA, but also one only by the traditional method. Just by thinking a bit about permutations and probabilities, one can see that this is not statistically significant.
The linear regression between number of snails and eDNA copies does not seem to have a uniform distribution of residuals, which is a prerequisite for applying linear regression. This results can therefore not be trusted either.
The water temperature mostly only takes three values - which makes other statistical tests more useful as well.
The authors talk about the problem of inhibition, but this has already been a problem in their previous paper; what has been done to tackle this problem since then?

Additional comments

In summary, I can only reject this paper. Sample size should be increased, replicates should be taken, followed by adequate statistical analysis. It would also be advantageous if the authors could trial different volumes of water, analyse the effect of up- and downstream sites, and include references to more recent eDNA work as well a short description of their previous work on mud snails.

Reviewer 2 ·

Basic reporting

No comment; the paper meets all requirements for basic reporting.

Experimental design

Paper generally meets requirements. Any comments on the methods are provided in the overall comments to the authors

Validity of the findings

No comment; findings and results make sense.

Additional comments

Overall:
This paper sought to compare results from a rapid eDNA sampler with on-site results compared to traditional sampling for the invasive New Zealand mud snail. The authors report that on-site eDNA site detection was higher than traditional sampling. The paper is clearly written, the results make sense and further add to ongoing eDNA research for conservation. Please see below I have provided more specific suggestions and comments.

Title: Minor issue, but consider replacing “for detecting” with “to detect”

Abstract:
L47: Minor suggestion to reword the end of the sentence to be “…compared to traditional visual survey methods.”
L50-51: Insert comma before “with five samples per site”


Introduction:
L66-72: Zebra mussels and red swamp crayfish are globally invasive, so not specific to Europe only.

L77: Revise “…, and then spreading” to read “and spread”

L80: Consider revising this sentence to reduce repetitiveness in use of “their”. Sentence could read, “…sampling methods due to small size, low population densities, patchy distribution, or complexity in use of habitat at different life stages.”

L85: Revise in-text citation to be Mächler et al. 2014

L92: Consider citation to Evans et al. 2017 as a more suitable study that directly examined the cost of eDNA vs. conventional sampling.

Evans NT, Shirey PD, Wieringa JG, Mahon AR, and GA Lamberti. 2017. Comparative cost and effort of fish distribution detection via environmental DNA analysis and electrofishing. Fisheries 42:90-99. https://doi.org/10.1080/03632415.2017.1276329

L98, L102: Minor formatting, revise in-text citations to use ;

L103: Minor comment to clarify that it is for freshwater or aquatic systems, given that terrestrial eDNA does not typically include collection of water. Maybe revise to read, “…there is not a standardized eDNA protocol for freshwater ecosystems, …”

L104-105: What does “with water collection stage showing to be a valuable step to increase detection rates” mean? Maybe revise this to read something like,

“Currently, there is not a standardized eDNA sampling protocol in freshwater ecosystems, but the overall sequence of steps includes water collection, preservation, and laboratory analysis. Further, research suggest that sampling decisions made during the water collection (i.e., volume of water) and handling step (i.e., on-site vs. lab filtration) may influence detection (add citation(s) here to support statements).”

Potential citations that authors may consider here:

Curtis AN, Larson ER, and MA Davis. 2021. Field storage of water samples affects measured environmental DNA concentration and detection. Limnology 22:1-4. https://doi.org/10.1007/s10201-020-00634-y

Mächler E, Deiner K, Spahn F, and F Altermatt. 2016. Fishing in the water: effect of sampled water volume on environmental dna-based detection of macroinvertebrates. Environmental Science Technology 50:305-312. https://doi.org/10.1021/acs.est.5b04188

Sepulveda AJ, Schabacker J, Smith S, Al-Chokhachy R, Luikart G, and SJ Amish. 2019. Improved detection of rare, endangered and invasive trout in using a new large-volume sampling method for eDNA capture. Environmental DNA 1:227-237. https://doi.org/10.1002/edn3.23

Materials & Methods:
L142-143: What does “positive detections” mean? Previous eDNA pilot study that confirmed eDNA detection here or that researchers had found mud snails here previously? I think it means snails have been recorded here previously, but use of positive detection made me think eDNA detection.

L149-152: Curious why samples were collected along the shoreline. Is this standard traditional sampling protocol?

L158-180: Consider moving this up to be the first section of the Materials and Methods to be before “Study Sites” section. This would allow the reader to have knowledge of the study organisms before reading the more specific information of the study sites and sampling.

L208-209: How many replicates per water sample were run? One qPCR replicate per water sample?

L210-211: Please provide information on the concentrations used to generate the standard curve. Additionally, it might be helpful to other researchers to know information about the gBlock, consider adding GenBank accession number or the sequence here or as supplemental.

Were plate blanks (NTCs) run?

Then if possible, provide estimates of the LOD, LOQ, R2, and % efficiency. Or are those values the same as reported in Thomas et al. 2018?

L231: I was confused here as to whether the sampling conducted was to confirm presence/absence of the snails or the eDNA. Please include a statement in this paragraph to state that snails were counted to get an estimate of site abundance, because this was not immediately clear to me.

L248: How many sites were removed?

Results:
L253-254: Clarify here what six sampling sites means. Does it mean stream or individual replicate along the 40 m transect? When I looked at the raw data that statement does not seem to be true. For example, neither eDNA or traditional sampling detected mud snails at the Columbia River/Kalama and only one sampling location had detection (eDNA only) at Capitol Lake. Then L255 states this, so consider deleting this sentence.

L256-257: I wonder if the eDNA was simply transported downstream from site 5 where the snail was at low abundance to the other sites (e.g. site 2) where eDNA was detected.

Discussion:
L279: Capitalize western (Western Washington) to match previous style

L282-283: Detection >1 snail/m2 is pretty good! Goldberg et al. 2013 can detect 1 snail/1.5 L of water, but in the field detection is at ~11 snails/m2. Also, as mentioned previously, this eDNA could simply be lower due to low abundance and transported downstream.

L299: What is mud snail DNA SQ? Starting quality of DNA? This abbreviation has not been referenced (also not explained in Figure 3A).

L301-302: Confused as to what “no formal analysis for this positive association” means. Isn’t this shown in Figure 3A?

L303: As a reader, I don’t understand what “which evaluated a limit of detection based on varying abundances of mud snails.” means here in context with the relationship between eDNA concentration and abundance of target organisms. Please revise to clarify this.

In this section about the relationship between species abundance and eDNA concentration, consider this reference which has found generally stronger relationships in laboratory studies and higher variability in field eDNA studies.

Yates MC, Fraser DJ, and AM Derry. 2019. Meta-analysis supports further refinement of eDNA for monitoring aquatic species-specific abundance in nature. Environmental DNA 1:5-13. https://doi.org/10.1002/edn3.7

L314-316: Is the relationship between temperature and eDNA maybe not significant because the methodology specifically targeted sampling during potential snail reproduction (as stated in L148-149) and maybe not due to degradation at relatively low temperature (14°C)? There are several eDNA papers that highlight the importance of reproduction (and increased temperature as a signal of reproduction) on eDNA detection. For example: Spear et al. 2015 or de Souza et al. 2016

de Souza LS, Godwin JC, Renshaw MA, and E Larson. 2016. Environmental DNA (eDNA) detection probability is influenced by seasonal activity of organisms. PLoS ONE 11:e0165273. https://doi.org/10.1371/journal.pone.0165273

Spear et al. 2015 is cited in-text at L310, but not in the reference list.

L318-319: The results do not show evidence of inhibition and from Figure 2 only one site was inhibited. Sample replication likely seems to be more important for detection, as stated in L324-327.

L341: In streams, transport, settling, and dilution can likely all influence detection.

References:
L499: Italicize Esox lucius

Tables/Figures:
Table 1 and 2: Nice work, these are very clear and easily to understand the results.

Figure 2: To me, presence/absence means detected/not-detected. But this figure is showing that and eDNA concentrations. Consider deleting the “Presence/Absence” from the first line of the figure legend.
Also, this figure is excellent! It very nicely displays the results of the study. Well done.

Figure 3: Consider moving the A, B, and C to the upper left of the figure.
Revise temperature units to read °C

Figure 4: The size of the figure legends and the yellow-red color here was difficult to read and is maybe not color-blind friendly. Consider either revising this or removing this figure because this information is mostly repetitive of Figure 2.

·

Basic reporting

This is a very well written, concise and interesting manuscript. The figures and tables were intuitive, and the written text was well polished.

Experimental design

The experimental design was sound - adequate replication and negative controls were included at all sites, and additional abiotic variables were measured which added an interesting component to the paper

Validity of the findings

The findings were discussed in relevant context, and the data provided in the supplemental file were sufficient to assess the validity of the study

Additional comments

I really enjoyed reading this paper, and was very impressed with the on-site collection, extraction and amplification method. I can see this being a useful method for monitoring a range of different species across different habitats, and the timely results could advise management decisions on the spot. A couple of the references could use some tidying up but other than that I'm happy to recommend the article be accepted in its current form

---

## Round 0.2 · Minor Revisions

Dear Jake and co-authors,

I have received another round of reviews. Reviewer 1 still has a problem with the use of the Fisher's exact test. I carefully looked at the rebuttal and reviewed the revised manuscript. My assessment is that this is a very valuable and good quality study that offers new insights into the use of in-field eDNA instruments for detecting the presence of a non-native mud snail from 6 freshwater streams in Canada. The use of the field-deployable Biomeme qPCR in this context is novel and exciting. A considerable effort went into field sampling and methods comparisons. The study is very well written, appropriately refers to previous work within the current context of the field, and provides sufficient details for others to replicate. All PeerJ editorial criteria are met, with only one issue relating to statistics. In my mind, this is fixable/addressable and does not qualify for a rejection.

There is indeed an issue with the lack of replicates (acknowledged by the authors) which makes the use of the Fisher's exact test and it interpretation problematic. Reviewer 3 has re-analyzed your data using a pseudo-replicated design, and concluded that you may want to consider focusing the paper on the detection sensitivity and possibly remove the contingency table and Fisher test. Reviewer 1 had a similar suggestion. Reviewer 2 had some excellent additional questions around Biomeme plate blanks and LOD/LOQ considerations.

Overall, the reviewers have provided you with excellent suggestions on how to address these remaining issues. I'll be looking forward to receiving your revised manuscript, along with the rebuttal.

With warm regards,
Xavier

Reviewer 1 ·

Basic reporting

The comments have been incorporated.

Experimental design

I agree that a final monitoring approach should include as many samples as possible; the proposed research here is however a benchmarking approach of the method so I think it would have been essential to include at least one/a few replicates at the very same location to understand how much the results change by just sampling one second later at the very same spot. Understanding this variability will be important to understand the efficiency of the approach – i.e. how many samples should be taken by a practitioner to confirm presence.

Thank you for adding a figure about the spatial distribution. Ideally, this figure would include the raw number of read counts (why and how were the thresholds chosen that are currently being shown?) and would show a direct comparison with the traditional field sampling results. As the statistical results comparing the traditional and eDNA approach are still not correct (see below), this would be a minimum requirement for accepting the paper.

I agree that trialling different volumes of water was not a necessity for this study, but again, it would have given practitioners important information about the sensitivity of the approach.

Validity of the findings

As briefly mentioned above, the statistical results are still not correct. Table 2 now just shows the number of sites that did or did not detect snails, with the eDNA results in one column and the traditional results in the other column. This does not show any comparison between the two methods at all and a Fisher's Exact test on these values is meaningless. We are interested in the site-dependent overlap between the two approaches.
The authors really have to think about what they want to show. If they did a Fisher's exact test on the site-dependent overlap as just suggested, the test would not be significant. This just shows that we don't have proof that the approaches are different - it does not automatically mean that the approaches are the same since the power of such a test always depends on sample size. If the authors want to show that the approaches mainly overlap, how about just showing the number of overlaps and then argue why some of the predictions did not overlap? Especially the cases where eDNA was negative and the traditional approach positive are worrisome.

Additional comments

Thank you for incorporating some of my suggestions. I am sorry to say that more needs to be done for me to accept this paper. The statistical results are still not correct and there is no clear message of the paper - is eDNA now better than traditional monitoring? (Some sites were only positive using the traditional approach.)

Reviewer 2 ·

Basic reporting

No comment. Paper appears to meet basic reporting standards.

Experimental design

No comment; see brief comments below

Validity of the findings

No comment

Additional comments

I commend the authors for doing a great job on thoughtfully and carefully revising their manuscript. I have some minor clarifying questions/comments below.

1. Still inconsistencies in "western Washington" and "Western Washington". Pick one and keep throughout. See Line 49 vs. 161

2. In the previous revision, I asked whether plate blanks were included and the response was that field blanks were included. Are machine/plate blanks not standard in field qPCR equipment? Field and plate blanks are standard now for eDNA research. I understand that the field blanks were confirmed negative and thus, it may be concluded that there was no contamination. Maybe I simply do not properly understand the Biomeme portable qPCR set-up and are reagents individually sealed to limit contamination?

3. On the limit of quantification: The authors cite to Thomas et al. 2018 S3, which lists LOQ at 50 copies and LOD at 10 copies. Did the authors then remove any samples that were <LOD/LOQ from analyses (Figure 3)? If not, what is the justification for not doing so?

4. L229-230: "...used nitrile gloves between each DNA extraction..." or changed gloves between extractions?

5. Literature Cited: Scientific names should be italicized. Peer J editorial staff may fix this for you.

·

Basic reporting

This paper is well-written and provides useful background information. The article is structured well. Raw data is provided as supplementary information.

Experimental design

The methods are explained in adequate detail. There is an issue with pseudo-replication in the survey design, since the 30 sites are not all independent of each other – 5 samples were taken at 10m intervals within each site – therefore I would argue that there were only 6 sites, and 5 replicates at each. The reps are not true technical reps (as pointed out by Reviewer 1 in the first round), but I think it makes more sense to interpret them as reps rather than independent sites. This is especially relevant here because DNA from the upstream sampling points could be carrying down and affecting the results of the downstream points.

I don’t think this is a disqualifying issue, but it does affect how the statistical analysis can be carried out:

1) The correct way to run the correlation analyses with a pseudo-replicated design is to take an average or median SQ value over the 5 reps at each site and correlate these with the conductivity/temperature readings and the visual counts. This would give only 6 data points rather than 30, but the log+1 of the mean SQ values still correlates nicely with the conductivity scores, and you could fit a trendline to the mean SQ vs the mean visual snail quantity – it’s probably not enough data for a p-value but the trend is evident.
2) The fisher exact test will also be affected since there are only 6 data points rather than 30. The corrected layout of the original table (I don't think th table was set out right in the MS) is:
eDNA_positive eDNA_negative
visual_positive 12 1
visual_negative 6 11

this gives a fisher’s exact p-value of 0.0024 – I’m pretty sure this is right - the R code I used is:
x <- matrix(c(12, 1, 6, 11), byrow = TRUE, nrow = 2)
fisher.test(x)

Since the design is pseudo-replicated the corrected layout of the new contingency table will be
eDNA_positive eDNA_negative
visual_positive 3 0
visual_negative 2 1

Unfortunately we can’t get a fisher’s statistic on this since the sample size is too small and snails were only visually observed at 3 of the 6 sites.

I’m actually not convinced that a contingency table makes a lot of sense here anyway, since the idea is to show that the eDNA is more sensitive, not that there is an association between visual counts and eDNA SQ values (this would seem intuitive). The detection sensitivity is covered off well in the previous analysis, so I’d consider getting rid of the contingency table and Fisher test.

Validity of the findings

Aside from the technical issues outlined above, the findings show the method to be sensitive and effective at picking up the target species in the field.

Additional comments

As a proof of concept this study gives a good introduction to the method and explains why in-field eDNA analysis is a useful and promising tool. I’m looking forward to hearing more as the method is refined and optimised and applied across a broad range of environments.

---

## Round 0.3 · accepted · Accept

Dear Jake and co-authors,

I have carefully reviewed your rebuttal and revised manuscript, and am happy to accept this manuscript for publication in PeerJ.

I'd like to thank all three reviewers for their time and efforts in improving the manuscript, and thank you for this great contribution to the field of eDNA!

With warm regards,
Xavier